# Testing the Predictive Validity and Construct of Pathological Video Game Use

**Christopher L. Groves** [1,†]**, Douglas Gentile** [1,*]**, Ryan L. Tapscott** [2,†] **and Paul J. Lynch** [3]

[1] Department of Psychology, Iowa State University, W112 Lagomarcino Hall, Ames, IA 50011, USA; E-Mail: cgroves@iastate.edu

[2] Department of Psychology, Grand View University, 1200 Grandview Ave., Des Moines, IA 50316, USA; E-Mail: rtapscott23@gmail.com

[3] Arizona Pain Specialists, 21803 N. Scottsdale Rd., Scottsdale, AZ 85255, USA; E-Mail: lynchmd@gmail.com

[†] These authors contributed equally to this work.

[*] Author to whom correspondence should be addressed; E-Mail: dgentile@iastate.edu; Tel.: +1-515-294-1472; Fax: +1-515-294-6424.

---

**Abstract:** Three studies assessed the construct of pathological video game use and tested its predictive validity. Replicating previous research, Study 1 produced evidence of convergent validity in 8th and 9th graders ($N = 607$) classified as pathological gamers. Study 2 replicated and extended the findings of Study 1 with college undergraduates ($N = 504$). Predictive validity was established in Study 3 by measuring cue reactivity to video games in college undergraduates ($N = 254$), such that pathological gamers were more emotionally reactive to and provided higher subjective appraisals of video games than non-pathological gamers and non-gamers. The three studies converged to show that pathological video game use seems similar to other addictions in its patterns of correlations with other constructs. Conceptual and definitional aspects of Internet Gaming Disorder are discussed.

**Keywords:** video games; addiction; internet gaming disorder

## 1. Introduction

Changes in technologies bring the potential for changes in users' thoughts, feelings, and behaviors [1]. Consequently, new media technologies, such as video games, often engender worries about potential problems associated with them [2]. Positive effects such as improved visual performance (e.g., [3]; increased pro-social behavior [4]; and improved surgical performance [5]) have been documented. Similarly, negative effects of video-gaming have been documented, such as disturbing school learning [6], increasing aggression ([7,8]), and decreasing pro-social behavior [9,10]. Although the general public colloquially speak of games being "addicting," historically there has been considerable debate about how to define internet/video-game addictions among clinicians and researchers interested in this phenomenon [11–15]. Recently, the American Psychological Association included Internet Gaming Disorder (IGD) in the appendices of the Diagnostic and Statistical Manual of Mental Disorders (DSM-5; [16]), calling for additional research on the topic.

The difficulty of defining problem video-game playing as an addiction mostly arises because video-gaming does not involve a chemical substance and because problems created by heavy use of video-games tend to be more benign as video-game playing is less likely to pose social threats through illegal activities, compared with drug addiction [17]. Although video-game playing does not involve intoxication or create external social threats as likely as substance use, research does show that those who excessively play video games report some addiction-like symptoms, including impairment in normal social and occupational or educational functioning, tolerance, withdrawal, relapse and the like [18–20]. This research evidence implies that even if problematic use of video game should not be labeled as "addiction", symptoms reported from heavy playing of video-games may be considered "pathological" enough to require clinical attention and interventions. Items based on the DSM criteria of behavioral addictions generally cover the domains of preoccupation, tolerance, loss of control, withdrawal, escape, and disruptions in schooling, family, and other social relationships.

From the experiences of clinical colleagues and increasing media reports, it appears that a number of gamers demonstrate symptoms of pathological computer and video game use [21,22]. Although some researchers have provided descriptive statistics about the pathological video-gamers [19,23–26], additional empirical evidence is still needed. Many of the prior studies (e.g., [23,25–27]) were conducted with a single sample of adolescents between 12 and 18. Hence, those studies may not generalize well to other groups, such as older adolescents who live a more independent life from parents in different educational settings. Studies are needed that test the construct in multiple samples.

Substance abuse and pathological gambling are often associated with antisocial personality disorder or aggressive behaviors (DSM-IV-TR; [28]). Given that the definition of pathological video-gaming in this study (as well as that of IGD in the DSM-5) shares the conceptual domains of pathology in the above disorders, it is expected that pathological video game players would show psychological and behavioral traits similar to pathological gambling or substance dependency or abuse, including antisocial or aggressive behaviors, hostility, or preference for violence, if the conception of it being a behavioral addiction is valid. That is, we might expect behavioral correlates that are similar to those that occur with other pathological behaviors.

Studies could also test the construct in other ways. For example, few studies have provided evidence of predictive validity, and the bulk of these studies have focused on predicting future addiction to video

game play [29]. To test predictive validity, it is useful to study pathological game players' actual responses to playing video games. Tolerance or withdrawal among alcohol or drug users is often considered partly conditioned with heightened responses to stimuli associated with the substance, such as the sight of drug paraphernalia. These heightened emotional and psychological responses have been called "cue reactivity." In a similar vein, pathological video game players may also show heightened reactivity to video-gaming. One case study demonstrated that a 22-year-old male self-identified video game addict had an increase in blood pressure from 113/75 to 140/69 when thinking about playing a game, and to 190/144 when playing a fighting game [30]. Another study used functional magnetic resonance imaging to examine the activation of neural structures associated with cue reactivity to images of video game play and smoking among individuals addicted to both nicotine and gaming [31]. Their findings implicated similar neural structures as active when participants were presented with either video game or smoking cues. Based on these findings, we can hypothesize that if pathological video gaming is similar to other addictions, pathological players would have heightened emotional responses to video games and rate the games to be more exciting, fun or stimulating, relative to non-pathological players.

The purpose of the present research was to test empirically the correlates and predictive validity of pathological video-gaming based on DSM-style criteria for pathological gambling. Three studies were conducted. Two correlational studies tested the construct of pathological video-gaming by comparing associations between it and other psychological traits, such as aggression and hostility, in samples of young adolescents in the 8th and 9th grade (Study 1) and older adolescents (Study 2). In Studies 1 and 2, we hypothesized that pathological gamers would show higher trait hostility, and higher antisocial and aggressive behaviors than non-pathological gamers. Study 3 was a quasi-experimental study that examined changes in emotional status before and after exposure to video games and appraisal of games played to test the predictive validity of pathological gaming, based on the concept of cue reactivity. The data in these three studies were collected prior to the publication of the DSM-5, although all three studies used scales designed to be similar to the DSM.

## 2. Study 1

### 2.1. Methods

#### 2.1.1. Participants

Data were collected from 607 young adolescents (46% female) in 8th-grade ($N = 496$) and 9th-grade ($N = 111$). Students were recruited from four Midwestern schools, including one urban private school ($N = 61$), two suburban public schools ($N = 350$), and one rural public school ($N = 196$). Students were recruited from mandatory classes within their schools, and participation was greater than 90% in all classes. The mean age of respondents was 14 years ($SD = 0.64$). Fifty-two percent of the participants were male. Eighty-seven percent of the respondents classified themselves as Caucasian, which is representative of the region of the country in which they reside. Participants were asked to report their frequency of playing video games on a scale from 1 (I never play video games) to 10 (almost every day). Non-gamers were identified as those who reported never playing video games ($n = 37$); remaining participants were classified as gamers ($n = 569$) with one missing value. Analyses are conducted on

gamers. All participants were treated in accordance with the ethical guidelines of the American Psychological Association (APA).

### 2.1.2. Procedure

All participants provided informed parental consent and individual assent. Each participant completed a version of the General Media Habits Questionnaire (GMHQ), a confidential survey that gathered data about students' media habits, attitudes, knowledge about video games, and pathological video gaming as well as school performance and demographic data [32,33]. The survey was pretested with 143 7th through 12th-grade students [34], and other data from this study are presented elsewhere [33]. Students also completed measures of personality trait aggression and hostile attribution bias. Classroom teachers were trained to administer the surveys during one class period. The students were instructed that videogames included any games played on computer, video game consoles (such as Nintendo or PlayStation), on hand-held game devices (such as Gameboy), or in video arcades.

### 2.1.3. Measures

Key variables in this study include pathological video gaming, personality trait hostility, hostile attribution bias, antisocial and aggressive behaviors, and preference for violence. Each variable was measured as follows.

#### Pathological Video Gaming

Seven items from the GMHQ assessed pathological gaming. These items were modifications of the DSM-IV criteria for pathological gambling, (similar to those used by Fisher [25], and Griffiths and Hunt [26]). Participants were considered to be pathological video gamers if they answered yes to at least four of the seven items. This cut point follows DSM-style criteria of requiring at least half of the diagnosable symptoms to be present. This approach of using a cut point of at least half of the symptoms was validated in a study of over 1100 youth [6]. The items are shown in Table 1. These data were collected prior to the DSM-5 criteria being finalized. Response options for the items were yes, sometimes, and no. However, in the analysis, sometimes was counted as yes. Reliability (Cronbach's alpha) was moderate ($\alpha = 0.61$) when conducted among gamers. This moderate reliability is predictable, given the small number of items and the fact that the scale was reduced to a dichotomous scale of yes and no.

#### Trait Hostility

Hostility was measured using the Cook and Medley Hostility Scale [35], a commonly used reliable instrument. Because the items for the Cook and Medley are taken from the MMPI, some were inappropriate for young adolescents. The instrument was modified by deleting seven items and changing the wording of some items to make them easier for 8th graders to understand, based on modifications made by Matthews and colleagues [36]. Scale reliability was acceptably high ($\alpha = 0.85$).

**Table 1.** Descriptive results for diagnostic items (Study 1).

| Items | % | Options Used | Number of Symptoms | % | n |
|---|---|---|---|---|---|
| - | - | - | 0 symptoms | 23% | 132 |
| 1. Do you ever play so much that it interferes with your homework? | 25% | Y/S | 1 symptom | 33% | 185 |
| 2. Do you feel restless if you cannot play video games? | 13% | Y/S | 2 symptoms | 19% | 106 |
| 3. Have you ever done poorly on a school assignment or test because you spent too much time playing video games? | 11% | Y/S | 3 symptoms | 14% | 77 |
| 4. Have you ever lied to family or friends about how much you play VGs? | 8% | Y/S | 4 symptoms | 6% | 33 |
| 5. Do you sometimes try to limit your own playing/ If yes, are you successful in limiting yourself? | 23% | N/S | 5 symptoms | 4% | 22 |
| 6. Have you ever played video games as a way to escape from problems? | 25% | Y/S | 6 symptoms | 2% | 10 |
| 7. After playing video games, do you often play again to try to get a higher score? | 54% | Y | 7 symptoms | 1% | 4 |

Note: Y=Yes; S=Sometimes; N=No. Percentage values were rounded to the nearest whole percentage point.

Hostile Attribution Bias

Hostile attribution bias is considered to be a social cognitive deficit, in which some people tend to attribute hostile motivations to others (e.g., [37]). The instrument included 10 scenarios, each describing an instance of provocation in which the intent of the provocateur is ambiguous [38,39]. Participants answer two questions following each story. The first presents four possible reasons for the peer's behavior, two of which indicate hostile intent and two of which reflect benign intent. The second question asks whether the provocateur(s) intended to be mean or not. The questions measure the participant's perception of hostility from the outside world, with higher scores indicating higher attribution of hostility to ambiguous events. Reliability for this scale was also high ($\alpha = 0.84$).

Antisocial Behavior

Antisocial behavior was measured by asking how often the participants had gotten into arguments with their parents, friends, and teachers in the past year. Responses were given on a 4-point Likert-type scale (ranging from "Less than monthly" to "Almost daily").

Physical Fights

Participants were asked if they had been in a physical fight in the last year. This question yielded a dichotomous response (yes/no).

Preference for Violence

Participants were also asked to indicate how much violence they prefer to have in their video games on a 10-point scale (1 = "no violence", 10 = "extreme violence"), and how much violence they prefer

to have in their video games compared to 2–3 years ago on a 5-point scale (1 = "a lot less", 5 = "a lot more").

*2.2. Results*

2.2.1. Descriptive Results

Table 1 displays the percentages of adolescents who display each of the symptoms of addiction, as well as the percentages of adolescents who display between zero and seven symptoms. Among our sample of young adolescent (8th–9th grade) gamers, 12% (*n* = 69) were considered pathological video-gamers as defined by reporting over half of the symptoms. As expected from the prevalence of pathological gambling and substance addictions, males were more likely to be pathological gamers than females. For young adolescents, the prevalence rates were similar to pathological gambling prevalence rates. For male gamers, 16% met the criteria for pathology (48 out of 306) while 7% of females met these criteria (19 out of 257; 2 participants had missing data for gender). Overall, participants spent a typical amount of time playing video games for this age group (*M* = 9.5 hours/week, *SD* = 12.1). However, pathological gamers averaged 21.6 hours/week (*SD* = 19.6), whereas non-pathological gamers averaged 7.9 hours/week (*SD* = 9.7).

2.2.2. Convergent Validity

It was predicted that pathological gamers would show patterns of correlations similar to those seen in other pathologies, including higher levels of antisocial behavior, hostility, and aggressive behavior. To this end, an analysis of covariance, controlling for gender, was conducted on these outcomes.

As displayed in the upper part of Table 2, pathological young adolescent gamers were more likely than non-pathological gamers to score higher on the Cook and Medley (1954) trait hostility scale, including scoring higher on each of the subscales. Pathological gamers were also more likely to display a hostile attribution bias and to report antisocial behaviors. They get into more arguments with their friends and parents, but were not significantly more likely to argue with teachers. Pathological gamers were more likely to report having been involved in physical fights in the previous year (49%, 33 out of 68 participants) than non- pathological gamers (33%, 163 out of 499 participants; $\chi^2$= 6.6, *df* = 1, *p* < 0.05, *n* = 567). They also reported significantly higher preference for violence in video games than non-pathological video-gamers. Due to the use of multiple tests, a conservative approach of using Bonferroni corrections was applied requiring a significance threshold of *p* < 0.004, which resulted in one test (arguments with parents) dropping below the significance threshold.

**Table 2.** Differences between pathological gamers vs. non-pathological gamers (Study 1: Young adolescents).

| Comorbidity with pathological status | *t* | *df* | Mean (*SD*) Path | Mean (*SD*) Non-Path | Mean diff. | Missing n Path | Non-Path |
|---|---|---|---|---|---|---|---|
| Hostile Attribution Bias | 2.46 * | 558 | 2.75 (1.12) | 2.34 (1.11) | 0.42 | 0 | 3 |
| Trait Hostility (Cook & Medley) | 5.74 *** | 551 | 24.63 (6.55) | 18.64 (7.27) | 5.99 | 3 | 8 |
| Cynicism subscale | 4.83 *** | 550 | 7.74 (2.06) | 6.03 (2.40) | 1.71 | 3 | 9 |
| Hostile attribution subscale | 5.99 *** | 544 | 5.70 (2.14) | 3.81 (2.24) | 1.88 | 3 | 15 |
| Hostile affect subscale | 3.87 *** | 536 | 3.22 (1.12) | 2.50 (1.35) | 0.72 | 5 | 21 |
| Aggressive responding subscale | 2.49 ** | 533 | 4.95 (1.79) | 4.16 (1.93) | 0.80 | 5 | 24 |
| Social avoidance subscale | 3.90 *** | 527 | 2.08 (1.31) | 1.37 (1.15) | 0.71 | 6 | 29 |
| Antisocial and Aggressive Behaviors | | | | | | | |
| In the past year, how often have you gotten into: | | | | | | | |
| arguments with your parents? | 2.41 * | 517 | 2.89 (0.96) | 2.62 (1.06) | 0.28 | 3 | 42 |
| arguments with your friends? | 2.86 ** | 481 | 2.16 (1.11) | 1.76 (0.92) | 0.40 | 8 | 73 |
| arguments with your teachers? | 1.28 | 408 | 1.94 (1.12) | 1.68 (1.06) | 0.27 | 16 | 139 |
| Have you been in a physical fight in past year | 6.66 * | 1 [a] | | | | 1 | 12 |
| Preference for violence | | | | | | | |
| On a scale of 1–10, how much violence do you like to have in video games? | 4.73 *** | 549 | 7.21 (2.54) | 5.14 (2.27) | 2.07 | 1 | 10 |

Note: *$p < 0.05$, **$p < 0.01$, ***$p < 0.001$. Path = pathological gamer group ($n = 69$), non-path = non-pathological gamer group ($n = 500$). [a] Because of the dichotomous nature of this variable, a chi-squared test is reported here.

## 3. Study 2

Study 2 was designed to replicate and extend Study 1 by using a different population (older adolescents who were undergraduates), by using some different measures, and by modifying the pathological video-gaming criteria. We again hypothesized that pathological gamers would show higher trait hostility, and higher antisocial and aggressive behaviors than non-pathological gamers. It is, therefore, a conceptual replication of Study 1, and would allow for greater generalizability of the data if the results were similar.

### 3.1. Methods

#### 3.1.1. Participants

Data were collected from 504 undergraduates (39% female) enrolled at a large Midwestern university. Students voluntarily participated and earned extra credit points for their introductory psychology classes. Gamer status was measured similarly to Study 1. Again, analyses exclude non-gamers ($n = 38$). Ninety percent of the respondents classified themselves as Caucasian, which is representative of the region and university population. All participants were treated in accordance with APA ethical guidelines.

#### 3.1.2. Procedure

Each participant completed three surveys: (1) scales from the adult version of the General Media Habits Questionnaire (GMHQ) that gathered descriptive data about students' video game habits including pathological video-gaming and demographic data; (2) a survey of aggressive and prosocial behaviors; and (3) a measure of trait hostility. Participants completed the surveys in a large group setting. The measures were embedded in a battery composed of various survey instruments submitted by several independent researchers.

#### 3.1.3. Measures

Pathological Video-gaming

Pathological video-gaming was measured similar to the approach described for young adolescents, except that nine items were used and participants were only able to answer "yes" and "no" without the option of "sometimes" to make the scale more similar to DSM (although again these data were collected prior to DSM-5 criteria being released) criteria. Participants were considered to be pathological gamers if they answered yes to five or more items, again based on DSM approaches to requiring at least half of symptoms to be present. Reliability was acceptable ($\alpha = 0.73$).

Trait Hostility

The Buss-Perry Aggression Questionnaire [40] was used to measure personality trait hostility ($\alpha = 0.91$). The Buss-Perry has four subscales: trait anger, hostility, verbal aggression, and overall physical aggression subscales ($\alpha$s = 0.80, 0.84, 0.78 and 0.84, respectively).

Aggressive and Prosocial Behaviors

Students completed the Social Interaction Survey [41], which measures self-reported use of six subtypes of aggressive behavior: proactive physical aggression (e.g., "I have threatened to physically harm other people in order to control them," 3 items, α = 0.68), reactive physical aggression (e.g., "When someone has angered or provoked me in some way, I have reacted by hitting that person," 3 items, α = 0.80), proactive relational aggression (5 items, α = 0.80), reactive relational aggression (5 items, α = 0.74), cross-gender relational aggression (5 items, α = 0.73), and one item measuring prosocial behavior.

Preference for Violence

Identical to the approach described in Study 1.

*3.2. Results*

3.2.1. Descriptive Results

Table 3 displays the percentages of older adolescent (undergraduate) gamers who displayed each of the symptoms of pathological video-gaming as well as the percentages who displayed between zero and eight symptoms (no respondent reported nine symptoms). In this version of the instrument, we dropped the item that had the highest endorsement in Study 1 ("After playing video games, do you often play again to try to get a higher score?"). This item was first included because "chasing" wins is indicative of a problem for pathological gamblers. This type of behavior is highly normative in video gaming, however, and seemed not to be a good indicator of pathological playing. Among this sample of gamers, 6% reported at least half of the symptoms (*n* = 30). For male undergraduate gamers, 9% met the criteria for pathology (29 out of 309) while 0.5% of females met these criteria (1 out of 195). Overall, participants spent a typical amount of time playing video games for this age group (*M* = 8.32 hours/week, *SD* = 10.38). However, pathological gamers averaged 17.6 hours/week (*SD* = 12.74), whereas non-pathological gamers averaged 8.19 hours/week (*SD* = 9.40).

**Table 3.** Descriptive results for diagnostic items (Study 2).

| Items | *%* | Options Used | Number of symptoms | *%* | n |
|---|---|---|---|---|---|
| - | - | - | 0 symptoms | 49% | 252 |
| 1. Do you ever play so much that it interferes with your homework? | 31% | Y | 1 symptom | 20% | 101 |
| 2. Do you become restless or irritable when attempting to cut down or stop playing video games? | 5% | Y | 2 symptoms | 13% | 68 |
| 3. Have you ever done poorly on a school assignment or test because you spent too much time playing video games? | 22% | Y | 3 symptoms | 7% | 38 |
| 4. Have you ever lied to family or friends about how much you play VGs? | 7% | Y | 4 symptoms | 5% | 23 |
| 5. Do you sometimes try to limit your own playing/If yes, are you successful in limiting yourself? | 8% | N/Y | 5 symptoms | 2.5% | 13 |

**Table 3.** *Cont.*

| Items | % | Options Used | Number of symptoms | % | n |
|---|---|---|---|---|---|
| 6. Have you played video games as a way of escaping from problems or bad feelings? | 31% | Y | 6 symptoms | 2.5% | 13 |
| 7. Do you need to spend more and more time and/or money on video games in order to achieve the desired excitement? | 6% | Y | 7 symptoms | 0.4% | 2 |
| 8. Over time, have you become more preoccupied with playing video games, studying video game playing, or planning the next opportunity to play? | 8% | Y | 8 symptoms | 0.4% | 2 |
| 9. Have you ever committed illegal/unsocial acts such as theft from family, friends, or elsewhere in order to get video games? | 2% | Y | 9 symptoms | 0% | 0 |

Note: Y=Yes; N=No. Percentage values were rounded to the nearest whole percentage point.

### 3.2.2. Convergent Validity

An analysis of covariance, controlling for gender, was used to examine the relations between pathological gaming status and several outcomes. Across measures, pathological older adolescents showed similar patterns of association with antisocial and aggressive traits and behaviors to those shown among younger adolescents (Table 4), although several of these relationships did not achieve statistical significance.

Pathological gamers scored higher than non-pathological gamers on the Buss-Perry trait hostility scale, though this difference was only marginally significant after controlling for gender. Pathological gamers scored higher on each of the subscales of the Buss-Perry although only the verbal aggression subscale reached the significance threshold and hostility was marginally significant. Pathological gamers were more likely to report being more proactively and reactively relationally aggressive. They were also more likely to report being proactively physically aggressive. The relationships between pathological status and reactive physical aggression, as well as helping behavior, were both in the predicted directions but neither relationship was significant after controlling for gender. Pathological gamers also reported significantly higher preference for violence in video games than did non-pathological video-gamers, as found in the sample of young adolescents. These results provide a strong conceptual replication of Study 1, given that the same pattern of results was found despite using a different age population and different measures of hostility and antisocial and aggressive behaviors. Again, a conservative approach to examining multiple tests involved the application of Bonferroni corrections, requiring a significance threshold of $p < 0.004$. This resulted in all tests except for reactive relational aggression dropping below the significance threshold.

**Table 4.** Differences between pathological gamers and. non-pathological gamers (Study 2: Older adolescents).

| Comorbidity with pathological status | *t* | *df* | Mean (*SD*) Path | Mean (*SD*) Non-Path | Mean Diff | Missing Path | n Non-path |
|---|---|---|---|---|---|---|---|
| *Trait Hostility (Buss-Perry)* | 1.83 [+] | 279 | 2.65 (0.80) | 2.26 (0.59) | 0.39 | 14 | 211 |
| Physical aggression subscale | 0.72 | 279 | 2.48 (0.76) | 2.01 (0.77) | 0.47 | 14 | 211 |
| Verbal aggression subscale | 1.98 * | 279 | 3.19 (1.25) | 2.68 (0.78) | 0.51 | 14 | 211 |
| Trait anger subscale | 1.38 | 279 | 2.43 (0.97) | 2.07 (0.72) | 0.36 | 14 | 211 |
| Hostility subscale | 1.74 [+] | 279 | 2.71 (0.97) | 2.31 (0.76) | 0.40 | 14 | 211 |
| *Antisocial and Aggressive Behaviors* | | | | | | | |
| Proactive physical aggression | 2.27 * | 278 | 5.31 (3.66) | 3.95 (1.70) | 1.36 | 14 | 212 |
| Reactive physical aggression | 0.78 | 275 | 5.88 (3.20) | 4.77 (2.63) | 1.11 | 14 | 215 |
| Proactive relational aggression | 2.53 * | 275 | 9.25 (3.61) | 7.29 (2.74) | 1.96 | 14 | 215 |
| Cross-gender relational aggression | 2.12 * | 241 | 10.79 (8.09) | 8.88 (4.42) | 1.91 | 16 | 247 |
| Prosocial behaviors | −1.05 | 273 | 5.31 (1.45) | 5.81 (1.37) | −0.50 | 14 | 217 |
| *Preference for violence* | | | | | | | |
| On a scale of 1–10, how much violence do you like to have in video games? | 1.77 [+] | 276 | 7.06 (2.21) | 5.31 (2.42) | 4.25 | 14 | 432 |

Note: [+]$p < 0.10$, *$p < 0.05$, **$p < 0.01$, ***$p < 0.001$. Path = pathological gamer group ($n = 30$), non-path = non-pathological gamer group ($n = 474$).

## 4. Study 3

Study 3 was designed to test the predictive validity of pathological gaming status. In Study 3, undergraduates played three video games and rated them on several dimensions as well reporting their emotional states prior to and immediately after playing the games. As described earlier, we hypothesized that pathological gamers should show heightened reactivity to video games if the measure of pathological video-gaming has predictive validity.

### 4.1. Methods

#### 4.1.1. Participants

The sample consisted of 254 undergraduates (48% female) at a large Midwestern university. Participants earned extra credit for their introductory psychology classes for participating in the study. Fifty-two percent of participants were male. Race information was not collected, but is likely to approximate 89% Caucasian as in Study 2. All participants were treated in accordance with APA ethical guidelines.

#### 4.1.2. Procedure

The study included aspects of both between-subjects and within-subjects designs. When participants arrived to the lab they completed an informed consent document and the GMHQ. Participants were randomly assigned to play three video games for 20 minutes each. Prior to playing each game participants completed the state emotion scale. Following each game, participants again completed the state emotion scale and a video game evaluation. Participants were debriefed and given credit. Pre-post game play emotional change scores were created for each participant. For this study, the critical tests were between pathological gamers, non-pathological gamers and non-gamers.

#### 4.1.3. Materials and Measures

*Video games*. Nineteen games of varying content (e.g., adventure, combat, sports, strategy, *etc.*) were used in the present study. Sixty-six percent of the games were considered by the experimenters as being non-violent and 34% were classified as violent (including intentional harm to other game characters). Each participant played three of these games, which were randomly selected. The selected three games were not all violent nor all non-violent.

Pathological Video-gaming

Each participant completed the General Media Habits Questionnaire (GMHQ). Pathological video-gaming was measured the same way as in Study 2. A total of 12 participants met the criteria for pathological gamer status (5%). Among the remaining participants, 203 were categorized as gamers (80%) and 39 were categorized as non-gamers (15%). Females were more likely to be non-gamers (87% female, $n = 34$) while males were more likely to be categorized as pathological gamers (17% female, $n = 2$). Males and females were more equally distributed in the gamer group (42% female, $n = 85$). Additionally, the test-retest reliability of the measure of pathological video-gaming was possible to evaluate in this study. Our undergraduate samples were free to sign up to participate in either or both

Study 2 and Study 3, and 47 students participated in both, allowing for an exploratory look at test-retest reliability of the pathological video-gaming markers. The dates on which these students participated in the experimental study were not recorded, hence the exact test-retest interval cannot be known. The experimental study (Study 3) ran throughout the semester, and the correlational study (Study 2) was conducted on one day about three-fourths into the semester. Assuming a random distribution of participation dates among students who participated in both studies, it is likely that the average test-retest interval was about a month. Test-retest reliability was high; the correlation ($r$) between the two sums of symptoms was .80 ($p < 0.001$). The correlation between the two dichotomous pathological video-gaming categories (pathological/not pathological) was also significant, at $\rho = 0.55$ ($p < 0.001$). Four participants moved from the non-pathological group to the pathological group in the experimental study. This may be due to three reasons. First, participants had volunteered to participate in a video game study and were given a more comprehensive video game habits survey than participants in Study 2. Therefore, participants answered questions all related to video games and were able to think more carefully about their habits than in Study 2, in which participants answered a battery of questionnaires of many types (that had been submitted by several researchers). Second, it is likely that students were more thoughtful and careful in the experimental study because they were tested individually in their own private rooms, whereas in the Study 2 they were in a room with hundreds of other students where there may be more emphasis on finishing quickly rather than carefully. Third, it may simply be the case that these students developed enough pathological symptoms to be classified as pathological players in the time period between their participation dates. Nonetheless, the test-retest reliability was high.

State Emotion

Participants completed a 32-item brief emotion checklist modified from the 132-item Multiple Affective Adjective Check List (MAACL; [42]) designed to measure state emotion (e.g., angry, happy, lonely, peaceful). Participants were instructed to check each of the adjectives that described their feelings at the current time. To control experiment-wise error, pre-post change scores following video game play were calculated for 13 of the 32 items. These 13 emotion terms were selected on an *a priori* basis as being the most theoretically relevant for pathological video gaming, and are displayed in Table 5.

Video Game Evaluation

Participants provided their impressions of the video games played on 14 dimensions (e.g., "The game was boring", "The game was exciting", "The game was violent", *etc.*) using a seven-point scale (1 = "strongly disagree" to 7 = "strongly agree"). The 14 dimensions are displayed in Table 6.

**Table 5.** Differences in emotional reactivity to playing video games (Study 3).

| Emotion Term | $\chi^2$ | df | Path Higher or Lower from Pre-Post | Pathological Gamer Emotion Change | | |
|---|---|---|---|---|---|---|
| | | | | −1 | *n* (expected *n*) 0 | 1 |
| Calm | 12.3 * | 4 | ↓ | 17 (12.4) | 14 (20.5) | 5 (3.1) |
| Peaceful | 13.9 ** | 4 | ↓ | 23 (13.4) | 13 (20.2) | 0 (2.4) |
| Pleasant | 9.5 * | 4 | ↓ | 18 (12.9) | 14 (20.0) | 4 (3.1) |
| Pleasant | 11.5 * | 4 | ↓ | 5 (1.6) | 19 (23.2) | 12 (11.2) |
| Irritated | 18.7 *** | 4 | ↓ | 6 (1.4) | 19 (21.4) | 11 (13.2) |
| Angry | 14.6 ** | 4 | ↑ | 0 (0.5) | 26 (31.9) | 10 (3.7) |
| Mad | 33.4 *** | 4 | ↑&↓ | 5 (0.7) | 27 (33.2) | 4 (2.1) |
| Happy | 7.7 + | 4 | ↑&↓ | 17 (13.1) | 13 (19.9) | 6 (3.0) |
| Energetic | 10.6 * | 4 | ↑ | 5 (5.8) | 23 (26.0) | 8 (4.2) |
| Powerful | 4.3 | 4 | = | 2 (2.8) | 28 (28.2) | 6 (5.0) |
| Lonely | 17.2 ** | 4 | ↓ | 10 (3.3) | 24 (31.5) | 2 (1.2) |
| Sad | 8.6 + | 4 | ↓ | 3 (1.4) | 33 (33.9) | 0 (0.7) |
| Unhappy | 10.2 * | 4 | ↓ | 5 (1.4) | 29 (31.6) | 2 (3.0) |

Note: $^+p < 0.10$, $^*p < 0.05$, $^{**}p < 0.01$, $^{***}p < 0.001$. Sample $N = 254$ in all tests, however, due to the simultaneous analysis of each of the three change scores, analyses were conducted on a sample of $N = 762$. Pathological gamer $n = 12$, non-pathological gamer $n = 203$, non-gamer $n = 39$.

**Table 6.** ANOVAs of video game evaluation ratings split by gamer type (Study 3).

| Rating statement | *F* | df | Path Higher or Lower than Other Groups | Means (*SD*) | | |
|---|---|---|---|---|---|---|
| | | | | Non-gamer | Gamer | Pathological Gamer |
| The game was entertaining | 10.6 *** | 2,751 | ↑ | 3.40 (1.95) | 3.89 (1.93) | 5.08 (1.52) |
| The game was exciting | 13.3 *** | 2,751 | ↑ | 2.87 (1.76) | 3.55 (1.87) | 4.61 (1.86) |
| The game was fun | 14.2 *** | 2,751 | ↑ | 3.05 (1.90) | 3.84 (1.85) | 4.78 (1.64) |
| The game was boring | 3.8 * | 2,751 | ↓ | 4.12 (2.05) | 3.67 (2.00) | 3.17 (1.72) |
| The game was absorbing | 15.0 *** | 2,750 | ↑ | 3.12 (1.86) | 3.54 (1.70) | 4.92 (1.52) |
| The game was arousing | 6.1 ** | 2,744 | ↑ | 2.45 (1.65) | 2.78 (1.66) | 3.56 (1.93) |
| The game was enjoyable | 10.7 *** | 2,749 | ↑ | 3.07 (1.92) | 3.69 (1.86) | 4.64 (1.57) |
| The game was involving | 12.5 *** | 2,750 | ↑ | 3.44 (1.96) | 4.00 (1.79) | 5.14 (1.38) |
| The game was stimulating | 14.9 *** | 2,749 | ↑ | 2.88 (1.79) | 3.54 (1.79) | 4.64 (1.62) |
| The game was addicting | 13.2 *** | 2,751 | ↑ | 2.42 (1.79) | 2.85 (1.91) | 4.28 (1.80) |
| The game was frustrating | 0.5 | 2,751 | = | 4.42 (1.94) | 4.57 (1.83) | 4.36 (2.05) |
| The game was difficult to play | 0.1 | 2,751 | = | 4.19 (1.90) | 4.15 (1.90) | 4.06 (1.70) |
| The game was action-packed | 1.1 | 2,751 | = | 3.41 (5.67) | 3.44 (2.07) | 4.17 (2.01) |
| The game was violent | 0.5 | 2,750 | = | 3.01 (2.46) | 3.27 (2.41) | 3.25 (2.34) |

Note: $^+p < 0.10$, $^*p < 0.05$, $^{**}p < 0.01$, $^{***}p < 0.001$. Sample $N = 254$ in all tests, however, due to the simultaneous analysis of each of the three change scores, analyses were conducted on a sample of $N = 762$. Pathological gamer $n = 12$, non-pathological gamer $n = 203$, non-gamer $n = 39$.

*4.2. Data Analysis*

The data reported here combine responses, resulting in each of the three responses being analyzed simultaneously (*i.e.*, each participants' change scores are analyzed at once, resulting in a total analysis sample of $N = 762$) with a chi-square test. This was done for three reasons. First, it confounds any order effects that may be present. From a theoretical perspective, pathological gamers should show higher reactivity regardless of order, whereas non-pathological gamers may have higher responses only to the first game if they were excited to get to play games as part of a study. This approach likely yields lower power to find effects if strong order effects are present. Second, it confounds any effects of the content of the games. Theoretically, violent video games should elicit higher responses for both pathological and non-pathological gamers. Therefore, it is important to eliminate any content-based effects. Third, because this study has a smaller sample, and because inclusion criteria were somewhat more strict compared to Study 1 (the exclusion of a "maybe" option) there were few pathological gamers ($n = 12$, 5%). Chi square analyses were performed here, rather than the more traditional repeated measures analysis, for two primary reasons. First, both groups suffer from the same independence assumption violation and therefore any error that results from the multiple measurements is applied evenly across groups, thus not prejudicing the results in any particular direction. Secondly, a violation of the assumption of independence ultimately increases the error term by not accounting for the variance that could normally be accounted for if we were conducting more typical repeated measures analyses with parametric data. Therefore, violation of this assumption should serve to prejudice the results against our hypotheses, as statistical power is lost.

*4.3. Results*

Theoretically, people who are physically or behaviorally addicted should show greater cue reactivity to stimuli that are related to their addiction than non-addicted controls. If pathological video-gaming is similar to other addictions, then pathological gamers should show greater responsiveness to games on several dimensions, including their appraisals of the games as well as changes in their emotional states. With regard to the game appraisals, we had directional hypotheses, predicting higher appraisals than non-pathological gamers on subjective dimensions (e.g., how fun, exciting, *etc.* the games were to play), but would not be different on objective dimensions (e.g., how violent the games were).

4.3.1. Emotional Responses to Playing Video Games

Prior to and following each game, participants selected emotions that they were currently feeling from a checklist. A pre-post difference was calculated by subtracting whether each adjective was marked before and after gameplay, allowing for tests of whether participants felt certain emotions more, less, or the same after each game. This produced three possible emotional change outcomes: those experiencing an emotional reduction were provided with a score of −1, those with no change received a score of 0, and those experiencing an emotional increase were provided with a score of 1. Because this was an experimental procedure, non-gamers also played the same games as the gamers, allowing for a comparison of responsiveness among three groups: pathological video-gamers, non-pathological video-gamers, and non-gamers. Because three total games were played, three emotional change scores were derived from

each participant (calculated by comparing the emotion scores prior to, and following each game). Chi-square tests were performed comparing three gamer types on the three possible emotion outcomes for several emotions. The results are shown in Table 5. Pathological video-gamers' changes in pre-post emotion are depicted in the table by arrows, showing the direction of change (e.g., less calm after playing the game, *etc.*). In addition, breakdowns of the emotional change scores among pathological gamers with the observed and expected distributions are provided in Table 5. Changes in almost all emotion terms were systematically related to gamer type. When the chi-square tables were examined, the significant differences were always related to the pathological video-gamers, and sometimes also to the non-gamers.

We hypothesized that pathological gamers would feel less calm and peaceful after playing a video game, as they should be highly reactive to them. We also expected that non-gamers would feel less calm and peaceful after playing, as they would not be used to playing them. This was the pattern found. Pathological gamers and non-gamers were both significantly more likely than would be expected to feel less calm, less peaceful, and less pleasant post-play. A different pattern was expected for agitation and irritation, such that pathological gamers were expected to feel less agitated and irritated, but non-gamers might be expected to feel more. This pattern was also found. Pathological video-gamers were significantly less agitated and irritated, but non-gamers felt more post-game agitation and irritation (Table 5).

Research has shown that playing violent games increases hostile feelings in the short term (e.g., [43,44]). Results in this study showed that pathological gamers were significantly more likely to feel increased anger post-play, and were over-represented for both increased and decreased post-play reports of feeling "mad" whereas non-gamers and non-pathological gamers were both more likely to feel the same pre- and post-play.

We hypothesized that pathological gamers would report higher post-play levels of feeling happy and energetic. The pattern was not this clear. Pathological gamers reported higher levels of feeling energetic post-play, as did non-gamers. However, pathological gamers as a group were more likely to feel *both* happier and less happy post-play whereas there was no systematic relation for non-gamers or non-pathological gamers.

Some researchers have suggested that a feeling of power may be a primary motivation for playing [45]. We predicted that pathological gamers might feel more powerful after playing, especially since they would be likely to be successful at playing whatever games we gave them to play. This prediction was not supported (Table 5).

Finally, we predicted that pathological gamers would feel less lonely, sad, and unhappy after playing. This pattern was found. Pathological gamers were more likely to feel less lonely, sad (marginally significant), and unhappy after playing a video game. There was no systematic relation for non-gamers or non-pathological gamers.

### 4.3.2. Game appraisals

After playing, participants rated each game on several dimensions. We hypothesized that when the dimension referred to a somewhat objective quality of the game (e.g., how violent or action-packed it is), that there would be no difference between gamer types, but when dimensions referred to a subjective personal appraisal of the game (e.g., how entertaining, stimulating, or fun it is), that pathological gamers would rate the games higher than non- pathological gamers or non-gamers. Analyses of Variance were conducted on each of the ratings, with gamer group as the between-groups factor (see Table 6; the arrows

indicate whether pathological gamers rated each statement higher or lower than the other two groups). Post-hoc comparisons showed that pathological gamers rated the games significantly higher than both non-gamers and non-pathological gamers for each of the subjective appraisals of the games. In addition, non- pathological gamers rated the games significantly higher than non-gamers for each of the subjective appraisals. There were only two exceptions to this pattern: the ratings on whether the game was "boring" and "frustrating." Non-gamers rated the games highest on boring, significantly different from both non-pathological and pathological gamers. Although pathological gamers rated the game less boring than non-pathological gamers, this comparison was not statistically significant. There were no significant differences on ratings of how frustrating the games were. As predicted, there were no significant differences between gaming groups on any of the objective qualities of the games.

## 5. General Discussion

In Studies 1 and 2, we tested pathological video-gaming with different populations and different measures. As there are established patterns of comorbidity for other substance and behavioral addictions, such as antisocial personality disorder, we predicted that pathological video-gaming should show similar correlations with hostility [46], aggressive behaviors [47], antisocial behaviors, and preference for violence in games. Each of these aspects was demonstrated. Compared with non-pathological gamers, pathological gamers scored higher on measures of trait hostility, engaged in higher levels of antisocial and aggressive behaviors, and had stronger preference for violence in video games. The stronger preference for violence in video games among pathological video-gaming could be argued to be evidence of tolerance [48,49]. In addition, the significant relationships between pathological video-gaming and aggressive and hostile traits imply the potential comorbidity of antisocial personality disorder for pathological video-gaming, although no clinical assessments were made in this study.

For the college sample, the results were very similar to those of younger adolescents despite being measured differently from Study 1. However, some of these relations did not meet the threshold for statistical significance in this older sample. Pathological gamers scored higher on a different personality trait hostility measure, and reported higher levels of antisocial and aggressive behaviors. Pathological gamers were also more likely than non-pathological gamers to report liking more violence in video games. This conceptual replication shows good evidence of the robustness of the construct, but suggests that the effect may be weaker among college students. Although the construct of pathological gaming has been examined in multiple age groups (e.g., [50,51]), far fewer publications have presented examinations of two separate age groups in the same report (e.g., [46]). Thus, the current work adds to these previous tests in supporting the generalizability of this construct across multiple age groups.

The prevalence of pathological video-gaming was lower in Study 2 than in Study 1 (6% and 12% of gamers, respectively). There are several reasons for the lower prevalence among older adolescents than younger ones. First, we modified our items to be stricter on four dimensions. The items were written to mirror the DSM-IV pathological gambling criteria more closely (because these data were collected prior to DSM-5 being available), whereas for the younger adolescents the items were based on these criteria, but were worded to be understandable to 8th graders. Similarly, two additional items were written to match the DSM more closely, one item was dropped and the diagnostic cut-point was raised for college students. Second, the options provided to respondents were made stricter for college students in Study 2, where most items only had yes/no options, whereas most items had yes/no/sometimes options for younger

adolescents and sometimes was grouped with "yes" for most items. Related to this point, we noted some cases of undergraduates marking "do not know" on several items rather than marking "yes". Several of these cases looked like they were pathological when considering their overall pattern of video game use, but were classified as non-pathological due to our strict criteria. The only potential surprise was that pathological gamers did not differ on their ratings of how frustrating the games were to play. It could be argued from a cue reactivity approach that pathological gamers should find the games more frustrating, but it could also be the case that pathological gamers were likely to be more proficient and therefore would find the games less frustrating. The lack of any significant effect does not shed any light on these two competing hypotheses.

Third, it may be that college students are less vulnerable to pathological video-gaming by virtue of their being a generally high-functioning group. Finally, it may be that developmental differences are implicated, such that younger adolescents are more vulnerable to pathological video-gaming, perhaps because they have fewer competing requirements for their time than college students. This study does not allow us to determine which, if any, of these accounts for the differences in prevalence rates. If there is some bias in our approach with older adolescents, it is possible that our percentages underestimate the prevalence in this age group. A national survey of American youth aged 8 to 18 put the prevalence at 8.5% of gamers [6]. Nevertheless, the significant relations between pathological video-gaming and hostility, aggression, and preference for violence in these two studies converge to suggest that pathological video-gaming as measured by a DSM-style checklist shows patterns similar to other addictions.

In Study 3, each participant played three games and provided information on their emotional states and judged several dimensions of the games. Theoretically, pathological gamers should show evidence of heightened reactivity. As hypothesized, pathological gamers reported greater changes in emotional states and rated their experiences of playing the games as more positive than non-pathological gamers and non-gamers.

Pathological gamers' pattern of emotional reactions to playing games was complex, however, and our interpretation should be viewed with caution until further research can replicate it. One interpretation is that pathological gamers reported less agitation and irritation after playing perhaps because it provided a "fix". Pathological gamers reported feeling less lonely, sad, and unhappy and more energetic after playing. Given that one criterion for addiction is that the player is motivated to play to escape from negative emotional states, these data support the idea that pathological gamers associated games with decreased negative feelings. However, they also reported feeling less calm, peaceful, and pleasant. This appears very similar to traditional cue reactivity, where presentation with an addiction-related stimulus increased symptoms of withdrawal and craving (e.g., [52,53]). In addition, the picture is unclear when considering happiness and anger. Some pathological gamers were more likely to be happier and less mad after playing, but some showed the opposite pattern. It is possible that these results were due in part to the checklist approach to measuring emotion, rather than by asking *how* mad participants felt. We would recommend that future research measuring emotional responses to games rate how much they feel each, rather than the dichotomous checklist used by the MAACL. If nothing else, this would be more sensitive to change. Note that this is not identical to traditional cue reactivity, as playing a video game for 20 minutes is more than a "cue". Nonetheless, the data supported the primary hypothesis that pathological gamers would be more emotionally reactive to playing video games.

When rating the gaming experience for each game, pathological gamers rated the games significantly more favorably than both non-gamers and non- pathological gamers. They considered the games to be more entertaining, exciting, fun, absorbing, arousing, enjoyable, involving, stimulating, and addicting than non-gamers and non- pathological gamers. They also rated the games as less boring than other participants. As predicted, they did not differ on ratings of more objective characteristics of the games, such as how violent, action-packed, or difficult the games were. It is for this reason that some researchers have included a "sometimes" category and have scored it as halfway between a "yes" and a "no" [6,46].

Several definitional issues remain to be studied. For example, we based our categorization on DSM-style criteria, where participants who answered yes to five or more of the diagnostic criteria were classified as pathological gamers and all others were classified as non-pathological. The studies reported here provide some evidence of validity for this dichotomous categorical approach using a cut-off point. However, the number of the diagnostic criteria a gamer presents to indicate levels of disorder could be equally useful. The fact that we found greater test-retest reliability for the number of criteria present than for whether the participants fell above or below our cut point suggests that additional studies should examine this issue, although we recognize the clinical value of introducing cut-off values for diagnostic purposes. Perhaps there are degrees of pathological use that would represent different challenges and would need to be treated differently. A related issue concerns how well checklist-style screening tools can differentiate highly engaged gamers from pathological gamers (e.g., [51,54]). The recent publication of the DSM-5 guidelines has led to what is becoming a very fruitful debate about which symptoms may best discriminate highly engaged from pathological levels (e.g., [31,55–59]). Indeed, this may currently be the largest challenge in the field. The studies presented here were not designed to provide a test of these questions, unfortunately, although we hope that the data may be useful for the discussion. For example, if a more conservative test were used (such as requiring the most problematic items, perhaps damage to grades and lying, to be endorsed for one to be considered pathological), the prevalence rates drop, but the overall pattern of correlations with other problem behaviors persists.

Our hypotheses were based on the theoretical position that most types of addictions should show similar patterns of correlations, such as with higher hostility and antisocial behaviors. Although this was largely confirmed, it is worth considering why that might be. There is no necessary reason why pathological gaming should predict aggressive behaviors. Our speculation is that this pattern is likely indicative of an underlying impulse-control problem, which is how we currently consider Internet Gaming Disorder. Future studies should test this hypothesis.

## 6. Limitations and Conclusions

Studies 1 and 2 are limited by their correlational nature, and we make no claims about whether pathological use is the result of, the cause of, or simply co-occur with the several comorbid factors we measured or could have measured. Because the purpose of these studies was to document patterns of relationships with pathological video-gaming, we did not report any analyses attempting to control for any of the factors other than gender. These data are possibly limited also by their date of collection (early-to-mid 2000s). It is unclear, however, exactly how the date of collection may limit them. Although games have changed dramatically in the past decade, this study focuses on the gamers rather than the games. Internet gaming disorder is defined by experiencing dysfunction due to games, not by the type of games played. As games have gotten more interactive, more engaging, and more available, it is

certainly possible that more gamers are having problems, but it is unclear why this would change the pattern of behaviors that are correlated with IGD. Nonetheless, this is an empirical question deserving of additional study.

The purpose of this study was to provide tests of the construct of pathological video gaming, not to validate an instrument. Nonetheless, the instrument used in Studies 2 and 3 shares many characteristics with the types of instruments being considered in response to the definition of Internet Gaming Disorder. For example, using Petry *et al.*'s (2014) [57] discussion of each of the nine symptoms, this scale includes eight of the nine symptoms: Pre-occupation (item 8), withdrawal (2), tolerance (7), unsuccessful attempts to stop or reduce (5), excessive gaming despite problems (1), deception (4), escape from a negative mood (6), and jeopardized or lost a relationship, job, or educational career opportunity (3). Our measure did not include an item to measure loss of interest in other hobbies or activities, although it did include a stricter and lower base rate item measuring theft. Thus, the data gathered here are likely to be useful for the ongoing debate on how these symptoms may be measured.

Although the literature dedicated to the topic of Internet Gaming Disorder is growing, it is still unclear at this point whether these are distinct problems of their own, or whether they are part of some broader pattern of pathologies. A large body of studies including outcomes of pathological use while controlling for various factors, such as comorbid pathologies, remain to be conducted. These will need to be conducted within theoretical frameworks of pathways to addiction. More research is needed regarding the etiology and course of pathological video-gaming. Future studies should continue to examine the developmental course of pathological video-gaming (e.g., [60]), establishing comorbidities of pathological use (e.g., depression, loneliness [61]), and especially what treatment options are most successful [62].

## Author Contributions

Christopher L. Groves was responsible for data analysis and writing. Douglas Gentile was responsible for theoretical development, study design, measurement, data collection, data management, analysis, and writing. Ryan L. Tapscott was responsible for data collection, data analysis, and writing. Paul J. Lynch was responsible for measurement and study design.

## Conflicts of Interest

The authors declare no conflict of interest.

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
