# Peer review of "Testing the Predictive Validity and Construct of Pathological Video Game Use"

_behavsci, doi:10.3390/bs5040602_

Round 1
Reviewer 1 Report
see attached pdf

Author Response
Main research interest of the manuscript at hand is to assess construct and predictive validity of pathological video games use. The manuscript certainly has major strengths such as combining three studies with different age groups. Unfortunately, there are some methodological concerns that have to be addressed. Below you will find my comments outlined according to the structure of the manuscript. I hope these suggestions will strengthen the final result.
Beforehand a couple of remarks, which will enhance the readability of the manuscript:
1. Please check again for when to use numbers expressed in numerals and numbers expressed in words (study 2).
2. There are a couple of unnecessary blank spaces in the manuscript.
3. Please unify sample descriptions (n and percentages)
4. There are various spellings of Coefficient alpha, i.e., Coefficient alpha, Cronbach´s alpha, α. Please unify.
5. Please unify the description as to how pathological was defined: “12.5 percent were considered pathological video-gamers (study 1)” vs. “5.8 percent reported at least half of the symptoms (study 2)”.
6. Please unify the structure of the study sections, e.g., methods is only to be found in study 1; results sections of study 1 and study 2, i.e., descriptive, convergent validity.
7. Study 3 has to be tightened and clarified (especially results section), to enhance reading and understanding.
8. Study 3, p. 9, state emotion needs a return (line break).
We have gone through the manuscript, making several edits in order to address each of the above points and believe that we’ve improved the consistency and clarity of the writing as a result.
Introduction
1. I am aware that research endeavors are sometimes overtaken by changes in circumstances, such as the changes introduced in DSM-5. The authors could even more offensively position their instrument with regard to these changes. Rather than writing “DSM-style criteria”, you could elaborate on commonalities and differences between the used instrument and the DSM-5 internet gaming disorder.
The reviewer makes a good point but we believe that our presentation of the exact items used in tables 1 and 3 help provide readers with the information needed to draw these conclusions. A primary reason for making this point in the manuscript is to justify our use of older diagnostic criteria.
Study 1
1. Descriptive results: ― On page 6 the authors state, that ”among our sample of young adolescent (8th-9th grade) gamers”. This implies that there were also non-gamers. One can only infer from the degrees of freedom in Table 2, that those were probably about 500, not accounting for missing values. Please provide numbers in the sample description. One can only speculate as to what “12.5 percent” refer to. ― Χ2-statistic: please also provide N. ― Please provide n of the subgroups pathological vs non-pathological. Please provide n´s for male and female pathological vs non-pathological as well. ― Table 1: Please provide n´s for symptom fulfillment and not only percentages.
2. Convergent validity: ― Table 2: Please provide descriptive statistics (M, SD, range, missing values) by subgroup (n), rather than only listing mean differences. ― Χ2-statistic: please also provide N.
― T-tests: As there is a multitude of t-Tests conducted, did the authors use Bonferroni correction? ― Given the uneven sample size of the two subgroups, the authors should consider using non-parametric methods. ― My biggest concern relates to the uneven distribution of gender in the two subgroups pathological vs. non-pathological. The various constructs used for assessing construct validity (trait hostility, hostile attribution bias, antisocial behavior, physical fights and preference for violence; see for example Krahé & Möller, 2010) are all more distinct among males. If gender is not controlled for, significant group differences may just be caused by a higher percentage of males in the group of pathological gamers. The authors could for example use matching techniques or restrict the analyses to males. I am aware that the authors state this limitation in the discussion section of the manuscript and refer to the correlational nature. However, as gender is highly confounded with group membership (pathological vs. non-pathological), it has to be controlled. The authors mention additional multivariate analyses in the discussion section, which are said to have demonstrated the same pattern of results. Please report these in the results section; at least those controlling for gender.
Krahé, B., & Möller, I. (2010). Longitudinal effects of media violence on aggression and empathy among german adolescents. Journal of Applied Developmental Psychology, 31(5), 401-409.
We’ve reported all requested analyses and changes above. Bonferroni corrections and gender breakdowns for both Studies 1 and 2 are provided in two new footnotes. We also decided to remove the χ2 analyses regarding the gender differences in pathological gamer status as these differences are fairly clear in the descriptive statistics provided and do not directly contribute to the primary findings reported in the paper. We also conducted some non-parametric tests, as recommended and found similar results. We decided to report the more traditional parametric tests, which are likely more familiar to readers.
Study 2
1. Again, the analyses seem to be restricted to gamers. Please provide numbers. One might only guess from the degrees of freedom in Table 4, that the percentage is probably larger in study 2 than in study 1.
2. Χ2-statistic: please also provide N.
3. Please provide n of the subgroups pathological vs non-pathological. Please provide n´s for male and female pathological vs non-pathological as well.
4. Table 3: Please provide n´s for symptom fulfillment and not only percentages.
5. Table 4: Please provide descriptive statistics (M, SD, range, missing values) by subgroup (n), rather than only listing mean differences.
6. Χ2-statistic: please also provide N. 7. T-tests: As a multitude of t-Tests were conducted, did the authors use Bonferroni correction?
8. Reconsider using non-parametric methods.
9. What I stated for study 1 is even a bigger concern for study 2: Given the uneven distribution of gender (95% males in the pathological gamers group vss. 58%), significant group differences may just be caused by a higher percentage of males in the group of pathological gamers. Gender differences must be accounted for.
These requested analyses and edits have been provided similarly to those conducted for Study 1. Please note that analyses for Study 2 were conducted from scratch and, while analyzed on the same dataset, were not perfectly replicated. Therefore, several values (e.g., study sample size) were modified to fit these more recent analyses. All values have been updated to fit these more recent analyses or left the same if results were identical.
Study 3
1. How many gamers were classified as pathological? 2. Please provide n of the subgroups pathological vs non-pathological vs non-gamers. Please provide n´s for male and female as well. 3. Data Analysis: I´m afraid, I could not understand how the mean Pre-Post game change scores were calculated: Did you average all pre and all post scores and then calculated the mean? Or did you calculate three change scores and then averaged across these three scores?
4. I did not find the results of the “pre and post differences across three types of gamers” (p. 10, line 9)? Did you use average pre scores and average post scores? Or did you use “PrePost game change scores”? This is essential for the analyses conducted. The analyses must control for pre group differences, otherwise results are difficult to interpret.
5. Table 5 and Table 6: Please provide pre and post descriptive statistics (M, SD, range, missing values) by subgroup (n), rather than only listing changes indicated by arrows
6. Χ2-statistic: please also provide N.
We’ve addressed all of these requested changes.
General Discussion
1. On page 12 you refer to significant correlations between pathological-video gaming and aggressive and hostile traits of study 1 and study 2. No correlations are reported in the results sections, but only group differences.
2. On page 13 you report 6 and 13 percent pathological gamers, in the results section it is 12.5 and 5.8. Please unify.
3. Please insert the subheading limitations.
We’ve changed “correlations” to “relationships” to more accurately reflect the types of analyses conducted. All percentages have been rounded and the subheading has been added.
Literature
1. DSM-IV-TR (2000): please correct. It is listed as DSM-III. Cite like DSM-5.
2. Please correct Rehbein et al. to (This error has been carried on numerously in other publications): Rehbein, F., Kleimann, M., & Mößle, T. (2010). Prevalence and Risk Factors of Video Game Dependency in Adolescence: Results of a German Nationwide Survey. Cyberpsychology, Behavior, and Social Networking, 13(3), 269-277. doi: 10.1089/cyber.2009.0227.
Both corrections have been provided. We wish to thank the reviewer for these valuable recommendations, which we believe have strengthened the manuscript.
Reviewer 2 Report
This paper reports on findings from three studies, aiming to assess the construct validity and predictive validity of pathological video game use. The paper is impressive in scope and provides a great amount of detail regarding the various hypothesized relationships between pathological video gaming and other correlates. This is an important part of establishing construct validity for a test. However, it is unclear whether the study truly accomplishes its goal of establishing construct validity and predictive validity, although it claims in the conclusions to have done so.
One issue, to begin with, is that it is unclear what the authors are aiming to establish the construct validity of? Normally, construct validity is established for a test as an indication of how well it captures what it intends to capture. In this study this is a bit unclear, as two different tests are used. If the study intends to provide construct validity of a test, then it needs to specify which test. However, I get the impression that what the authors aim for is something like construct validity in broader terms (and thus pertaining to both tests?) - essentially theory validation? This should be specified somewhere, as it is quite an important distinction.
Moving on, a study of construct validity in my mind is chiefly a contribution to theory building – it is a matter of establishing the extent to which inferences made on the basis of a proposed theory might be valid reflections of the intended construct. With this in mind, I feel that the present study would benefit from considering, peripherally, some alternative theories than the one proposed. At present, only one interpretation is offered for each hypothesis and, typically, the most straightforward interpretation is presented. While this is perhaps reasonable given the aim of the study, it is also problematic because it ignores the amount of critical work that has emerged on this topic in recent years. In doing so, it fails to convincingly explain why the proposed theory should be used over other, alternative, theories. One consequence of this is that the results are left hanging; they are not very well connected to some of the recent critical developments in this area. Although the results presented are impressive and many hypotheses supported, it is unclear how precisely these results might contribute to the continued study of pathological video gaming, given that the theoretical foundation for these hypotheses have been subjected to some critiques which in this paper remain unmentioned and unanswered. While the paper provides an excellent discussion around the conceptual dilemmas of labeling problematic gaming as an ‘addiction’ (page 2), and justifies well its choice of the term ‘pathological video gaming’, the same level of reflexivity and critical awareness is also needed in regards to the theory and interpretation of results.
Below I will outline a number of points for the authors to consider:
Page 2 line 34: “Although some researchers have provided descriptive statistics about the pathological video-gamers (Chiu, Lee, & Huang, 2004; Durkee et al., 2012; Fisher, 1994 ; Griffiths, 2000; Griffiths & Hunt, 1998), empirical evidence for the construct validity of pathological video-gaming is still needed. Many of the prior studies (e.g., Chiu et al., 2004; Fisher, 1994; Griffiths & Hunt, 1998) were conducted with a single sample of adolescents between 12 and 18. Hence, the construct validity demonstrated in those studies may not generalize to other groups”
I am a bit concerned about the references to old studies here – such as Fisher and Griffiths & Hunt. Video gaming has undergone rather extreme changes since 1998 and, arguably, even since 2004. It might be useful for the sake of your argument to find more recent studies to cite.
Second, while it is certainly true that construct validity of pathological video-gaming is needed, no study up until this point has presented adequate evidence of construct validity according to the standards as advocated by Cronbach & Meehl (1955) or Loevinger (1957). I would therefore carefully suggest that the authors re-write this section as to not make it sound like prior studies have provided evidence of construct validity – at best, some studies have provided partial evidence of construct validity, but even this is inadequate because of some serious conceptual and methodological issues. More on this will follow later as it is perhaps the main issue of the study – the term construct validity is used a bit too loosely. Construct validity really needs to be properly defined at the outset of the study – in this review my understanding draws mostly on Cronbach & Meehl and Loevinger as above, but recent movements have argued for alternative definitions.
Page 3 line 1: “Furthermore, available evidence suggestive of the construct validity of pathological video-gaming is rarely built on known psychological or behavioral issues associated with pathological video-gaming. “
I am curious about this statement. In my understanding, almost all available evidence suggestive of the construct validity of pathological video-gaming, and a majority of the studies on this topic, are built precisely on known psychological or behavioral issues associated with pathological video-gaming. I would argue that studies of correlations between psychological or psychiatric issues and some problem-gaming-construct are over-represented in this field of study. I wonder what the authors would argue that the current body of evidence is built upon?
Page 3 line 3: “According to the DSM-IV-TR (2000), substance dependence or abuse or pathological gambling is often associated with antisocial personality disorder or aggressive behaviors. Given that the definition of pathological video-gaming in this study shares the similar conceptual domains of pathology in the above disorders, it is expected that pathological video game players would show psychological and behavioral traits similar to pathological gambling or substance dependency or abuse, including antisocial or aggressive behaviors, hostility, or preference for violence, if the construct is valid.”
Here is a key sentence for this study – excellently phrased. It makes clear the theoretical foundation for the study, which is that pathological video-gaming is considered to bear conceptual similarity to substance abuse or pathological gambling. This means that any subsequent confirmatory validation only pertains to this particular theory (if I understood the authors purpose correctly), not necessarily to the latent construct for which this theory is only one possible interpretation – this distinction should be made clearer in a study of construct validity and will be facilitated by the definition of construct validity as suggested above.
Page 3 line 11: “To test predictive validity, it is necessary to study pathological game players’ actual responses to playing video games. Tolerance or withdrawal among alcohol or drug users is often considered partly conditioned with heightened responses to stimuli associated with the substance, such as the sight of drug paraphernalia. These heightened emotional and psychological responses have been called “cue reactivity”. In a similar vein, pathological video game players may also show heightened reactivity to video-gaming. […] Based on these findings, we can hypothesize that if pathological video gaming is similar to other addictions, pathological players would have heightened emotional responses to video games and rate the games to be more exciting, fun or stimulating, relative to non-pathological players.”
This is indeed a good point and represents the sort of hypothesis-driven theory testing that a validation study should contain. Based on the theoretical definition of pathological video-gaming we might assume behavioral correlates similar to other pathological behaviors – such findings would indeed strengthen the construct validity of this proposed interpretation.
However, as Smith (2005) has argued: “It is important to remember that Cronbach and Meehl’s (1955) emphasis was not on recording a few successfully predicted correlations. Because construct validation involved basic theory testing, they emphasized principles for making inferences about the meaning of test scores or experimental outcomes. Since their early contribution, methodologists have periodically sought to remind investigators of this crucial perspective.” And: “[This] would help prevent us from viewing construct validation as collecting a series of stamps: a content validity correlation, a criterion-related validity correlation, and so on.”.
I believe these quotes by Smith once again emphasizes the need for some nuance as to what this study might contribute. His paper can be found here:
Smith, G. (2005). On Construct Validity: Issues of Method and Measurement. Psychological Assessment, 17(4), 396-408.
Moving on to discuss the specific studies:
As regards Study 1 – I wonder when this data was collected? Some of the data seems to have been used as long ago as 2004 (Page 4, line 10)? As mentioned previously, the gaming industry is an industry that has undergone some dramatic changes since early 2000 and data collected back then might not be very representative in today’s environment. Attitudes and access to video games have changed and increased, often for the better according to representative data, which could have some implications for a study where self-report measures were used to indicate a ‘less desirable’ behavior like pathological gaming. This needs to be reported and, if appropriate, discussed in the limitations section (which is currently missing?).
Page 4 line 20: “Seven items from the GMHQ assessed pathological gaming. These items were modifications of the DSM-IV criteria for pathological gambling, (similar to those used by Fisher, 1994 and Griffiths & Hunt, 1998). Participants were considered to be pathological video gamers if they answered yes to at least four of the seven items. This cut point follows DSM-style criteria of requiring at least half of the diagnosable symptoms to be present.“
While this cut off is certainly in line with DSM and common practice, it is also potentially problematic because a majority of the criteria used do not actually capture problematic engagement. Rather, they try to mirror behavioral manifestations as observed in relation to substance abuse or pathological gambling. This has been a point of contention in the field for some time, as can be derived from the following 4 studies, and makes the operationalization problematic:
Kardefelt-Winther, D. A critical account of DSM-5 criteria for internet gaming disorder. Addict Res Theory 2014; 1-6. DOI:10.3109/16066359.2014.935350
Charlton, J. P., & Danforth, I. D. (2007). Distinguishing addiction and high engagement in the context of online game playing. Computers in Human Behavior, 23(3), 1531-1548.
Griffiths, M. D., van Rooij, A., Kardefelt-Winther, D., Starcevic, V., Király, O., Pallesen, S., et al. (2015). Working towards an international consensus on criteria for assessing Internet Gaming Disorder: A critical commentary on Petry et al. (2014). Addiction, in press.
King D, Delfabbro P. Is preoccupation an oversimplication? A call to examine cognitive factors underlying internet gaming disorder. Addiction 2014; 109: 1566-1570.
As Charlton & Danforth’s (2007) has discussed, tests of pathological gaming fail to distinguish between engaged gamers and pathological gamers. Therefore, it should be acknowledged that the items used to assess pathological gaming in this study might actually capture people who are just very engaged gamers. This is particularly true of the older measures used here, compared to those in DSM-5, even though some problems remain.
This is a point which the authors might wish to reflect upon, in particular since it has some impact on how Table 1 is interpreted. Arguably, only 2 items out of 7 reflect a problematic outcome (Item 1 and 3). The rest might as well reflect high engagement, or that the family might not be appreciative of gaming as a hobby. For example, the criteria with the highest percentage of ‘Yes’ responses is “After playing video games, do you often play again to try to get a higher score?“. This illustrates some of the problems with the criteria used here. Playing again to get a higher score is the basic function of many video games, in particular older games which this study might be capturing. In fact, the entire point of early arcade and computer games was to get a higher score – Pac man for example revolved almost entirely around this concept. While this indicator (of Tolerance?) to ‘keep using more and more’ is a valid indicator of problematic behavior in the context of substances, due to the hazardous and often illegal nature of substance use, the context of video game playing is not perfectly comparable with substance use. Therefore some criteria used here might not be relevant indicators of problematic gaming behavior, which illustrates a weakness of the theoretical approach and operationalization used in this study – this should be discussed.
This problem of has been thoroughly documented in the 4 papers mentioned above. An engaged gamer might thus easily score 4 or more on the instrument used here and will be labeled as a pathological gamer, while in fact being an enthusiastic gamer. It would be particularly important for this paper to discuss how the items used here correspond to the distinction of problematic VS high-engagement gamers, taking Charlton & Danforth (2007) as a point of departure, and reflect on what the subsequent problems in this study might be and the impact on evidence for construct validity. This is important as this theoretical distinction is, perhaps, one of the biggest challenges for the field today – a valid construct must clarify this distinction.
Furthermore, given the concerns mentioned here around the scale used to measure pathological video-gaming, it might be worthwhile to consider Items 1 and 3 as items that must be present, in addition to a number of additional criteria, for pathology to be identified. This would perhaps reduce the prevalence figures to be more in line with those found in recent research using DSM-5 measurements (i.e., between 0.4-2%).
Finally, as regards Study 1, all hypotheses regarding convergent validity (co-occurrence of psychological problems similar to those seen in other pathologies) were supported. These findings make sense and are easily interpreted if viewed from the theoretical perspective of pathological video-gaming as conceptually similar to substance abuse or pathological gambling. However, I’m missing some further reflection around ‘why’ pathological gamers would express more hostility and be more anti-social. Indeed, this is notoriously difficult to say given the pathological interpretation that the authors have chosen, where the most commonly heard argument for a ‘why’ is that “it happens because it also happens in relation to substance abuse”, which to me feels somewhat inadequate. Why would pathological gamers, according to a theory of pathology, be more hostile or anti-social? This has not been discussed in the literature review. Construct validity demands a well articulated theory - such a theory should be able to express why the construct, theoretically, should/should not be related to other constructs.
Finally, I am a bit stumped about the finding that pathological gamers play on average 21h a week – it seems questionable that 3h a day could possible cause any severe, long-term real life problems? In the truly problematic, clinical cases that I encounter the children barely leave their room and play 12+h every day. Interestingly, the reported SD is 18.2 which might suggest some heavy outliers? It would be interesting to see how this mean value might differ if the authors try a different cut-off with the ‘must be endorsed’ criteria as suggested previously. While time itself is of course unlikely to be a direct predictor of pathological video-gaming, 3h a day seems a bit too low to merit clinical relevance. Some reflections of what this might mean would be interesting, in particular since it was largely similar in Study 2. In fact, come to think of it, the reported descriptives for Study 2 are identical – perhaps a mistake in reporting the findings here?
Study 2
I note that in Study 2, the item which received the highest amount of ‘Yes’ responses in Study 1 is now removed - “After playing video games, do you often play again to try to get a higher score?“. I think this is worth highlighting, since an item 59% of the sample endorsed is no longer included in the second study which, after all, aims to replicate the first. Does it have an impact on the overall conclusions? Might be worth discussing if this impacts the convergent validity.
Study 3
The hypothesis here is that pathological gamers should show heightened reactivity to video games if the measure of pathological video-gaming has predictive validity. This is predicated on the theoretical definition of pathological video-gaming which states that the response to gaming should be similar to conditioned responses of drug users when presented with stimuli associated with the substance. Problematically, there is no way to distinguish engaged gaming from harmful gaming (because the test lacks construct validity!). Considering the problems expressed earlier relating to the measurement of pathological gaming, the sample used here might just as well be constituted partly of engaged gamers. This could just as well explain why all of the hypotheses as were supported: 1) engaged gamers are highly reactive to games because they like them 2) engaged gamers are less agitated and irritated after playing, because playing is fun 3) engaged gamers report higher post-play happiness/less sadness/less unhappiness because, again, playing is fun. Therefore, I would like to ask of the authors to elaborate on how certain they can be that their sample of gamers and their heightened ‘cue reactivity’ might not simply be explained by the fact that these people really enjoy games, are happy to see them, are stimulated by playing and absolutely delighted to get course credits for playing! In other words, how do you know that the predictive validity you propose here points to the correct construct, given the lack of construct validity for the test used? As two studies is not enough as evidence for construct validity, this is a problem that needs to be discussed. If the construct validity for your test is uncertain, as it arguably is due to heavy disagreements in the field about its operationalization (see e.g., below paper), then it follows that your measures will also be uncertain as adequate indicators for this concept.
Griffiths, M. D., van Rooij, A., Kardefelt-Winther, D., Starcevic, V., Király, O., Pallesen, S., et al. (2015). Working towards an international consensus on criteria for assessing Internet Gaming Disorder: A critical commentary on Petry et al. (2014). Addiction, in press.
This means that it is not a given that the inferences you make are supported, which is not to say that the results are not important, but the framing of results needs to be made with this in mind. While this is a natural part of scientific progress, in particular for theory-building (e.g., Smith, 2005), the kind of experimental study proposed here rests on some questionable assumptions until the construct validity of the latent variable is solid – this is currently not the case. One particular reason for why this is not the case in the field of gaming addiction studies is that the item-pool used to create measures intended to capture the latent construct are always constrained to our understanding of substance use disorder. This means that our interpretation is always constrained by the data we have collected, which means that no conclusions about construct validity can be drawn from this data. The following texts elaborate on this in some detail:
Loevinger, J. (1957). Objective tests as instruments of psychological theory. Psychological Reports, 3, 635-694.
Cronbach, J., & Meehl ,P. (1955) Construct validity in psychological tests. Psychol. Bull, 52, 281-302.
The real implications of the comments above for a study such as this one, is that the authors need to be a bit more careful in their claims of having established construct validity – this is a very complex endeavor that requires a great deal of iterative hypothesis-testing. The three studies presented here are part of such an endeavor, despite their somewhat problematic measurement of pathological video-gaming, but it is not enough. The reason I highlight this is because claims of construct validity might lead other researchers to believe that we know much more about the present construct than we actually do and lead them to use measurements that are actually not properly validated.
Therefore, moving towards the conclusions, I suggest that there needs to be some reappraisal of the statement on page 10, line 34: “In Studies 1 and 2, we tested the construct validity of pathological video-gaming with different populations and different measures.”
As mentioned previously, you certainly assessed part of the construct validity as pertaining to your theoretical definition – this is surely important work. However, because there is more to construct validity than testing a few hypotheses (Smith, 2005), this statement is currently a bit misleading.
Similarly, page 11 line 9: “This conceptual replication shows good evidence of the robustness of the construct.” - It shows some robustness of the proposed test as a reflection of your theory, but it is not enough to support the later claim on page 12, line 27: “these studies provide evidence of construct validity.”. These are serious claims that have implications for how other researchers might chose to proceed in their own work – given the shortcomings in particular of the dependent variable, it might be reasonable to be a bit more prudent regarding claims of providing evidence for construct validity – again, a lot more work is needed. This study is a good start.
I think this sentence on page 13 line 7 is revealing: “Because the purpose of these studies was to document patterns of correlations with pathological video-gaming, we did not report any analyses attempting to control for any of the factors when predicting others.”.
This is how I would suggest framing the whole study - as exploring patterns of correlations with pathological video-gaming, rather than proposing to establish construct validity. If you wish to make claims about construct validity there needs to be a solid grounding in the literature on psychological assessment which specifies in detail what precisely is needed to establish construct validity and how far this study goes in meeting those needs – in these studies you focus on the matter of convergent validity and predictive validity, which certainly provides some evidence of construct validity, but it is another matter to fully establish that a theory or a test is a valid representation of the construct (i.e. has construct validity). I think it would be important to provide some clarity regarding this matter and some nuance as to what it is you actually accomplish here.
Finally, given the recent inclusion of Internet Gaming Disorder in DSM-5 and considering some of the recent critiques against its operationalization, I’m wondering what we might do with the construct as presented here? How can we move forward with it? Should we revert to this understanding of the behavior and discard the proposed changes in DSM-5? Or do the changes in DSM-5 in fact build on this construct presented here? Some reflections on how we might use the findings presented in this study would be helpful.
Author Response
Reviewer #2:
This paper reports on findings from three studies, aiming to assess the construct validity and predictive validity of pathological video game use. The paper is impressive in scope and provides a great amount of detail regarding the various hypothesized relationships between pathological video gaming and other correlates. This is an important part of establishing construct validity for a test. However, it is unclear whether the study truly accomplishes its goal of establishing construct validity and predictive validity, although it claims in the conclusions to have done so.
One issue, to begin with, is that it is unclear what the authors are aiming to establish the construct validity of? Normally, construct validity is established for a test as an indication of how well it captures what it intends to capture. In this study this is a bit unclear, as two different tests are used. If the study intends to provide construct validity of a test, then it needs to specify which test. However, I get the impression that what the authors aim for is something like construct validity in broader terms (and thus pertaining to both tests?) - essentially theory validation? This should be specified somewhere, as it is quite an important distinction.
Moving on, a study of construct validity in my mind is chiefly a contribution to theory building – it is a matter of establishing the extent to which inferences made on the basis of a proposed theory might be valid reflections of the intended construct. With this in mind, I feel that the present study would benefit from considering, peripherally, some alternative theories than the one proposed. At present, only one interpretation is offered for each hypothesis and, typically, the most straightforward interpretation is presented. While this is perhaps reasonable given the aim of the study, it is also problematic because it ignores the amount of critical work that has emerged on this topic in recent years. In doing so, it fails to convincingly explain why the proposed theory should be used over other, alternative, theories. One consequence of this is that the results are left hanging; they are not very well connected to some of the recent critical developments in this area. Although the results presented are impressive and many hypotheses supported, it is unclear how precisely these results might contribute to the continued study of pathological video gaming, given that the theoretical foundation for these hypotheses have been subjected to some critiques which in this paper remain unmentioned and unanswered. While the paper provides an excellent discussion around the conceptual dilemmas of labeling problematic gaming as an ‘addiction’ (page 2), and justifies well its choice of the term ‘pathological video gaming’, the same level of reflexivity and critical awareness is also needed in regards to the theory and interpretation of results.
We agree with the reviewer that the term construct validity was not used in the most appropriate way in our earlier draft. We did not, however, feel that this manuscript was the best place to lay out all of the nuances of the issue, especially since our data could not cleanly resolve the questions. Therefore we have removed the term, and instead tried to be clearer that we are providing some tests of the construct of pathological gaming.
Page 2 line 34: “Although some researchers have provided descriptive statistics about the pathological video-gamers (Chiu, Lee, & Huang, 2004; Durkee et al., 2012; Fisher, 1994 ; Griffiths, 2000; Griffiths & Hunt, 1998), empirical evidence for the construct validity of pathological video-gaming is still needed. Many of the prior studies (e.g., Chiu et al., 2004; Fisher, 1994; Griffiths & Hunt, 1998) were conducted with a single sample of adolescents between 12 and 18. Hence, the construct validity demonstrated in those studies may not generalize to other groups”
I am a bit concerned about the references to old studies here – such as Fisher and Griffiths & Hunt. Video gaming has undergone rather extreme changes since 1998 and, arguably, even since 2004. It might be useful for the sake of your argument to find more recent studies to cite.
We have added additional references.
Second, while it is certainly true that construct validity of pathological video-gaming is needed, no study up until this point has presented adequate evidence of construct validity according to the standards as advocated by Cronbach & Meehl (1955) or Loevinger (1957). I would therefore carefully suggest that the authors re-write this section as to not make it sound like prior studies have provided evidence of construct validity – at best, some studies have provided partial evidence of construct validity, but even this is inadequate because of some serious conceptual and methodological issues. More on this will follow later as it is perhaps the main issue of the study – the term construct validity is used a bit too loosely. Construct validity really needs to be properly defined at the outset of the study – in this review my understanding draws mostly on Cronbach & Meehl and Loevinger as above, but recent movements have argued for alternative definitions.
As noted, we have removed the framing of this study being to test construct validity.
Page 3 line 1: “Furthermore, available evidence suggestive of the construct validity of pathological video-gaming is rarely built on known psychological or behavioral issues associated with pathological video-gaming. “
I am curious about this statement. In my understanding, almost all available evidence suggestive of the construct validity of pathological video-gaming, and a majority of the studies on this topic, are built precisely on known psychological or behavioral issues associated with pathological video-gaming. I would argue that studies of correlations between psychological or psychiatric issues and some problem-gaming-construct are over-represented in this field of study. I wonder what the authors would argue that the current body of evidence is built upon?
We agree that this is a curious statement, and wonder what we meant by it when we originally wrote it. It has therefore been removed.
Page 3 line 3: “According to the DSM-IV-TR (2000), substance dependence or abuse or pathological gambling is often associated with antisocial personality disorder or aggressive behaviors. Given that the definition of pathological video-gaming in this study shares the similar conceptual domains of pathology in the above disorders, it is expected that pathological video game players would show psychological and behavioral traits similar to pathological gambling or substance dependency or abuse, including antisocial or aggressive behaviors, hostility, or preference for violence, if the construct is valid.”
Here is a key sentence for this study – excellently phrased. It makes clear the theoretical foundation for the study, which is that pathological video-gaming is considered to bear conceptual similarity to substance abuse or pathological gambling. This means that any subsequent confirmatory validation only pertains to this particular theory (if I understood the authors purpose correctly), not necessarily to the latent construct for which this theory is only one possible interpretation – this distinction should be made clearer in a study of construct validity and will be facilitated by the definition of construct validity as suggested above.
We have removed the construct validity frame, and have also added additional discussion of alternative theoretical frames through which our data could be viewed in the general discussion.
Page 3 line 11: “To test predictive validity, it is necessary to study pathological game players’ actual responses to playing video games. Tolerance or withdrawal among alcohol or drug users is often considered partly conditioned with heightened responses to stimuli associated with the substance, such as the sight of drug paraphernalia. These heightened emotional and psychological responses have been called “cue reactivity”. In a similar vein, pathological video game players may also show heightened reactivity to video-gaming. […] Based on these findings, we can hypothesize that if pathological video gaming is similar to other addictions, pathological players would have heightened emotional responses to video games and rate the games to be more exciting, fun or stimulating, relative to non-pathological players.”
This is indeed a good point and represents the sort of hypothesis-driven theory testing that a validation study should contain. Based on the theoretical definition of pathological video-gaming we might assume behavioral correlates similar to other pathological behaviors – such findings would indeed strengthen the construct validity of this proposed interpretation.
However, as Smith (2005) has argued: “It is important to remember that Cronbach and Meehl’s (1955) emphasis was not on recording a few successfully predicted correlations. Because construct validation involved basic theory testing, they emphasized principles for making inferences about the meaning of test scores or experimental outcomes. Since their early contribution, methodologists have periodically sought to remind investigators of this crucial perspective.” And: “[This] would help prevent us from viewing construct validation as collecting a series of stamps: a content validity correlation, a criterion-related validity correlation, and so on.”.
I believe these quotes by Smith once again emphasizes the need for some nuance as to what this study might contribute. His paper can be found here:
Smith, G. (2005). On Construct Validity: Issues of Method and Measurement. Psychological Assessment, 17(4), 396-408.
We have tried to make this point clearer.
Moving on to discuss the specific studies:
As regards Study 1 – I wonder when this data was collected? Some of the data seems to have been used as long ago as 2004 (Page 4, line 10)? As mentioned previously, the gaming industry is an industry that has undergone some dramatic changes since early 2000 and data collected back then might not be very representative in today’s environment. Attitudes and access to video games have changed and increased, often for the better according to representative data, which could have some implications for a study where self-report measures were used to indicate a ‘less desirable’ behavior like pathological gaming. This needs to be reported and, if appropriate, discussed in the limitations section (which is currently missing?).
The reviewer is correct, the data were collected quite a while ago. This has now been made more explicit and a discussion of the limitations has been expanded to include this issue.
Page 4 line 20: “Seven items from the GMHQ assessed pathological gaming. These items were modifications of the DSM-IV criteria for pathological gambling, (similar to those used by Fisher, 1994 and Griffiths & Hunt, 1998). Participants were considered to be pathological video gamers if they answered yes to at least four of the seven items. This cut point follows DSM-style criteria of requiring at least half of the diagnosable symptoms to be present.“
While this cut off is certainly in line with DSM and common practice, it is also potentially problematic because a majority of the criteria used do not actually capture problematic engagement. Rather, they try to mirror behavioral manifestations as observed in relation to substance abuse or pathological gambling. This has been a point of contention in the field for some time, as can be derived from the following 4 studies, and makes the operationalization problematic:
<span style="f